# A Large Mode Area Parabolic-Profile Core Fiber with Modified Segmented in Cladding

**Song Yang** [1], **Wentao Zhang** [1], **Yulai She** [2,*], **Hao Du** [1] **and San Tu** [3]

1. Guangxi Key Laboratory of Optoelectronic Information Processing,
   Guilin University of Electronic Technology, Guilin 541004, China
2. School of Mechanical and Electrical Engineering, Guilin University of Electronic Technology,
   Guilin 541004, China
3. Guangxi Key Laboratory of Nuclear Physics and Technology, Guangxi Normal University,
   Guilin 541004, China
* Correspondence: yulai_she@guet.edu.cn

**Abstract:** In this paper, a novel fiber with a parabolic-profile core and eight segmented trenches in cladding is designed. The designed fiber consists of the segmented trench of low refractive index in cladding and parabolic-profile of high refractive index in the core. The proposed fiber has good bending resistance. The eight segmented trenches in the cladding can decrease the refractive index of cladding to increase the difference index between the core and cladding to limit fundamental mode (FM) loss. The proposed fiber can offer an effective single mode (SM) operation with a large mode area (LMA) of 952 $\mu m^2$ at the small bending radius (R = 10 cm). In addition, the fiber is also suitable under an 18 W/m heat load. The proposed fiber achieves good bending performance, which can be suitable for the compact high-power lasers.

**Keywords:** large mode area; single-mode operation; bending loss; fiber design

## 1. Introduction

Compared with conventional gas and solid-state lasers, high-power fiber lasers have a series of advantages, such as lightweight, high beam quality, corrosion resistance, and low threshold [1–4]. In recent years, high-power fiber lasers have received much attention from scholars, and their power levels have exceeded the 10-kW class [5,6]. However, nonlinear effects [7,8] and optical damage become more severe with a further increase in power. An effective method commonly used to solve these problems is to use large mode field area fibers [9].

Large mode field area fibers will be bent and sometimes curled in practice due to space constraints. So, large mode field area fibers should be allowed to operate at a moderate bending radius. Moreover, large mode field area fibers must work in a single mode operation to ensure beam propagation quality in high-power lasers. However, the challenge is increasing the mode field area of the fiber and decreasing the loss of FM to maintain single-mode operation [10,11].

In recent years, people have proposed several fiber structures to achieve the requirement of single-mode operation in a large mode field area with good bending robustness, e.g., photonic crystal fibers [12–18], low numerical aperture (NA) step-index core fibers [19–22], Bragg fibers [23,24], multi-trench fibers [25–29], and leakage channel fiber [30–33] etc. However, these fibers have some problems. When photonic crystal fibers are spliced, the air holes in the fibers collapse and the stacking technology makes manufacturing more difficult. Low-NA step-index fibers achieve single-mode operation by reducing the cutoff wavelength. Although increasing the core diameter helps to reduce the NA, it is complicated to produce fibers with a NA of less than 0.06 during practical preparation [34]. Bragg fiber has shown good performance in enlarging the mode field area. However,

the high refractive index ring in the fiber creates unwanted coupling with the core. The mode field area of the multi-trench fiber becomes smaller in the case of bending. Shaoshuo Ma proposed an improved multi-trench fiber structure with bend-resistance and large mode field characteristics, which can obtain a mode field area of 840 μm$^2$ for effective single-mode operation at a bend radius of 15 cm [28]. For leakage channel fibers where intermodal coupling can be an issue due to a much increased mode density, Dong has shown that fundamental-mode operation is possible in all glass leakage channel fibers with core diameter beyond 100 μm [30].

This paper proposes a novel fiber with a parabolic-profile core and eight segmented-trenches in cladding (PPC-SCF). The influence rule of each parameter on bending performance of PPC-SCF has been summarized in detail. In order to show the improvements, the proposed and traditional step-index fibers are compared under the same parameters. After modulation by a parabolic-profile core and segmented trenches. The proposed fiber can offer an effective SM operation with a LMA of 952 μm$^2$ at the small bending radius (R = 10 cm). In addition, the fiber is also suitable under an 18 W/m heat load. Our design shows great potential in SM operation, mode-area enlargement, and bending resistance.

## 2. Theory and Structure

Figure 1 shows the transverse cross-section of the proposed fiber. The grey region represents the silica material. The refractive index is $n_1$. The blue region represents segmented trenches. The refractive index is $n_2$ and angular width is $\theta_2$. The angle width between two segmented trenches is $\theta_1$. The yellow region represents core in fiber and the refractive index is $n_3$. The dn is differences in refractive index between refractive index of silica material and low refractive index. The dn1 is differences in refractive index between highest refractive index in the core and refractive index of silica material. The core radius is a, and the cladding radius is 72.5 μm. c is the thickness of optical fiber and perfect matching layer (PML), as shown in Figure 1a. φ is the bending orientation. The duty cycle of the segmentation is $\gamma = \theta_2/(\theta_1 + \theta_2)$. Figure 1b,c show the refractive index along X and X′ directions, respectively. The incident wavelength is 1.064 μm. The bending radius R is defaulted to 10 cm. Bending orientation θ is fixed as 0°.

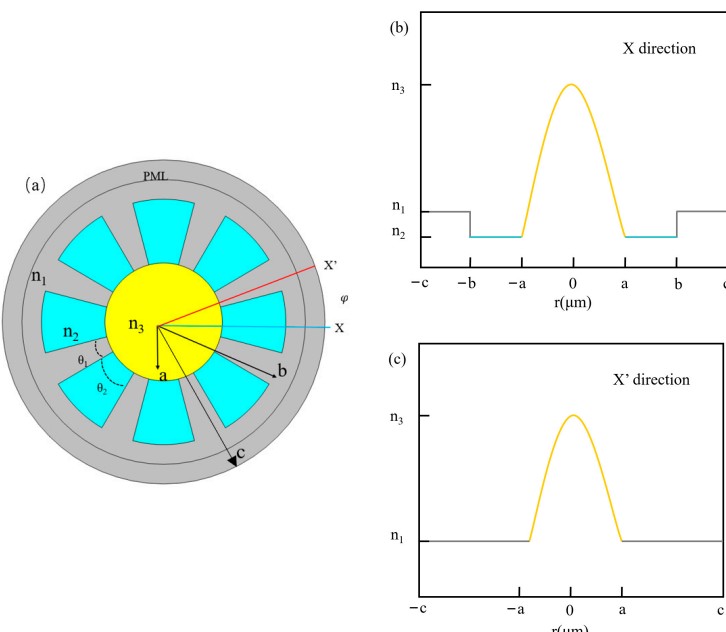

**Figure 1.** (**a**) Schematic cross-section of the proposed structure. (**b**) Schematic of refractive index profile of proposed fiber along X direction. (**c**) Schematic of refractive index profile of proposed fiber along X′ direction.

We use COMSOL Multiphysics commercial software based on the finite element method to perform numerical simulations. Adding a circular PML outside the cladding for calculating the bending loss, the loss of each mode in the fiber is constant when the thickness of the PML is greater than 5 μm. In this paper, the thickness of the PML is set to 10 μm. The Mesh generation of the proposed fiber as shown in Figure 2. The average unit mass is 0.88, and there are 41,425 degrees of freedom.

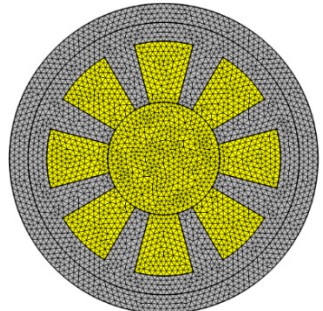

**Figure 2.** Mesh generation of the proposed fiber.

The proposed fiber has the RI in the core distribution of:

$$n(x,y) = n_3 - (n_3 - n_1)(x^2 + y^2)/a^2 \tag{1}$$

To calculate the bending loss of the proposed fiber, the refractive index of bending fiber can be expressed equivalently to the straight fiber structure. The refractive index equivalence equation is as follows [35]:

$$n_{eq}(r,\theta) = n(r) + (1 + \frac{x\cos\varphi + y\cos\varphi}{\rho * R}) \tag{2}$$

where $n_{eq}$ is equivalent refractive index of bending fiber. $n(r)$ is the refractive index of the straight fiber. $r$ is the coordinate axes of fiber. ρ is the elastic optical coefficient, θ is the bending azimuthal angle, and R is the bend radius. In this article, the value of ρ is set as 1.25 according to the literature [36].

The bending loss can be calculated using the following equation [35]:

$$Loss(dB/m) = \frac{40\pi}{\ln(10)\lambda} Im(n_{eq}) \tag{3}$$

where λ is the incident wavelength, λ = 1064 nm in this paper.

The effective mode area ($A_{eff}$) reflects the magnitude of the power density inside the fiber. Improving the $A_{eff}$ can reduce the nonlinear effects. $A_{eff}$ is calculated as [37,38]:

$$A_{eff} = \frac{\left(\iint |E(x,y)^2|d_x d_y\right)^2}{\iint |E(x,y)|^4 d_x d_y} \tag{4}$$

where $E(x,y)$ represents the fiber transverse electric field component.

Because of the quantum defect, the rare earth doped fiber will generate heat load during use. The heat load will influence the equivalent refractive index of the proposed fiber. The relationship between heat load and refractive index can be calculated as [39]:

$$n_T(q,r) = \begin{cases} n_1 + \frac{qr_c^2\beta}{4k_{s_i}}\left[1 + \frac{2k_{si}}{h_1} + 2ln\left(\frac{b}{r_c}\right) + 2\frac{k_{si}}{k_{ac}}ln\left(\frac{c}{b}\right) - \left(\frac{r}{r_c^2}\right)^2\right] & 0 \le r \le a \\ n_2 + \frac{qr_c^2\beta}{2k_{s_i}}\left[\frac{2k_{si}}{h_1} + ln\left(\frac{b}{r_c}\right) + 2\frac{k_{si}}{k_{ac}}ln\left(\frac{c}{b}\right)\right] & \text{cladding } a \le r \le b \\ n_3 + \frac{qr_c^2\beta}{2k_{s_i}}\left[\frac{2k_{si}}{h_1} + ln\left(\frac{b}{r_c}\right) + 2\frac{k_{si}}{k_{ac}}ln\left(\frac{c}{b}\right)\right] & \text{Low-index in cladding } a \le r \le b \end{cases} \tag{5}$$

where $c$ is the radius of the fiber coating, $\beta$ is the thermal optical coefficient, while $k_{si}$ and $k_{ac}$ are the thermal conductivity for the silica and coating material. $h_1$ is the convective coefficient between the coat and air. Parameters of value are shown by Table 1 [39].

**Table 1.** Parameter of value.

| Parameter | Value |
|-----------|-------|
| c | 82.5 µm |
| β | $3 \times 10^{-5}$/K |
| $k_{si}$ | 1.38 W/(m·K) |
| $k_c$ | 0.2 W/(m·K) |
| $h_1$ | 80 W/(m²·K) |

$q$ is the heat load density, which can be expressed as [39]:

$$q(z) = \frac{Q(z)}{\pi a^2} \tag{6}$$

where $Q(z)$ is the heat load.

## 3. Discussion of Proposed Fiber

First, in order to demonstrate the superiority of the PPC fiber to resist bending, the traditional step-index fiber is analyzed to compare it with the proposed fiber. Figure 3 shows the electric field distribution of the proposed fiber and step-index fiber with the structure parameters a = 30 µm, b = 62.5 µm, dn = dn1 = 0.0006, $\gamma$ = 0.65, $\lambda$ = 1.064 µm, and R = 10 cm. Figure 3a,b plots the electric mode field of the $LP_{01}$ and $LP_{11v}$ with proposed fiber. Figure 3c,d shows the electric field of the $LP_{01}$ and $LP_{11v}$ with the step-index fiber. Figure 3a plots that the electric mode field of the parabolic core fiber, the mode of $LP_{01}$ is a circle. However, the $LP_{01}$ mode field of the step-index fiber is clearly compressed near the edge of the core, as shown in Figure 3b. The electric field lines of $LP_{11v}$ mode have been radiated into the cladding, resulting in a large loss. The $LP_{01}$ and $LP_{11v}$ loss of the parabolic core layer is 0.022 dB/m and 1.963 dB/m, respectively. The effective touch-field area is 751.8481 µm². Figure 3d shows that the electric field lines of $LP_{11v}$ mode were restricted to the core layer. The bending loss of $LP_{11v}$ is small. The $LP_{01}$ and $LP_{11v}$ loss of step-core fiber are 0.022 dB/m and 0.021 dB/m, respectively. The effective field area is 788.89 µm². The proposed fiber can realize the single mode operation compared with step-index core fiber at the same bending radius.

To verify the bending performance of the PPC fiber and step-index fiber, the normalized electric field distribution has been analyzed, as shown in Figure 4. The normalized electric field of PPC fiber (red dot) is closer to the center compared with the step-index fiber (blue dot) at bending radius R = 10 cm. Therefore, the PPC fiber has better bending performance than step-index fiber.

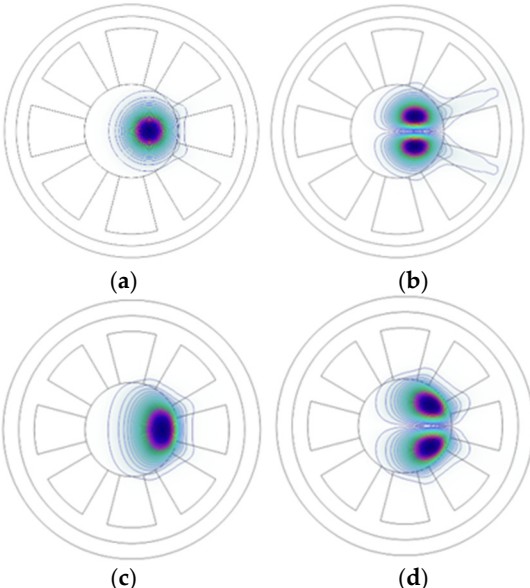

**Figure 3.** Transverse electric mode field of (**a**) $LP_{01}$, (**b**) $LP_{11v}$ in PPC-fiber and (**c**) $LP_{01}$, (**d**) $LP_{11v}$ in step-index fiber with R = 10 cm.

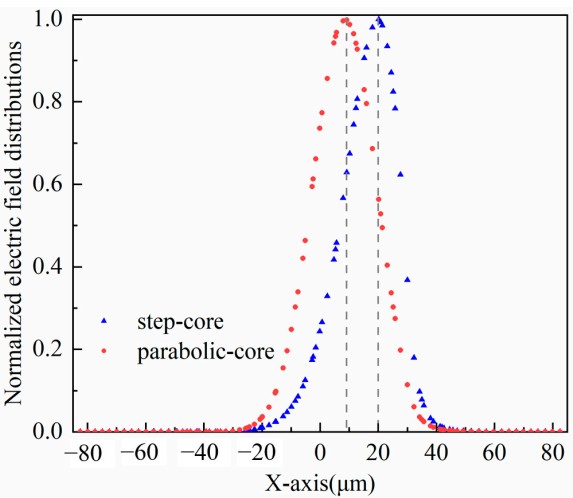

**Figure 4.** Normalized electric field distribution of $LP_{01}$ along the radial direction with the bending radius is 10 cm.

### 3.1. Effects of Core Radius

Firstly, the influence of core radius on fiber bending performance is analyzed. Figure 5 shows the bending loss, loss ratio, and effective mode area as a function of a. The parameters are: $\lambda$ = 1064 nm, b = 62.5 μm, dn = dn1 = 0.0006, R = 10 cm, $\gamma$ = 0.65, $\theta$ = 0°. Figure 5a shows that the bending loss of $LP_{01}$ decreases at first and then increases with core radius increasing. The $LP_{01}$ loss decreases from 0.0491 dB/m to 0.0136 dB/m with core radius ranging from 15 to 24 μm. With the core radius increasing from 24 to 35, the bending loss of $LP_{01}$ increases from 0.0136 dB/m to 0.052 dB/m. When the core radius is small, the core has weak ability to limit light, which results in a high bending loss. When the core radius increases, the light capacity rises, and the light bending loss decreases. As the core radius continues to increase, it becomes increasingly sensitive to the bending radius. Therefore, the bending loss will increase.

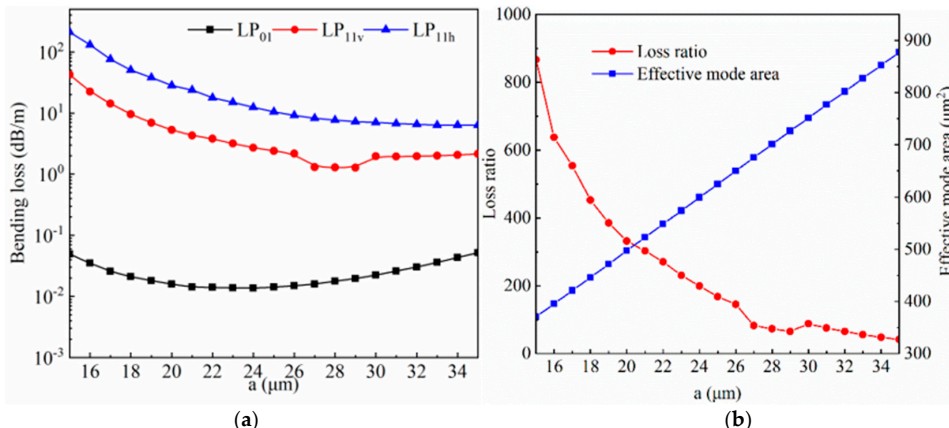

**Figure 5.** (**a**) The bending loss of $LP_{01}$ and $LP_{11}$ in the core (**b**) loss ratio and effective mode area as the function of a.

Figure 5b shows the loss ratio between the lowest high order modes (HOMs) and the highest FM and the effective mode area. The mode area of $LP_{01}$ keeps increasing with core radius increasing. When core radius is 15 μm, the effective mode field area is 370.385 μm$^2$. When a increases to 35 μm, the effective mode field area can reach 877.56 μm$^2$. The overall trend of the loss ratio is down. With core radius ranging from 16 μm to 35 μm, the loss ratio reduces from 866 to 41.

### 3.2. Effects of Number of Segments

The number of segments with low-index cladding will affect the refractive index difference between the core and the cladding layer. The gap between the segments will also affect light energy leakage. Therefore, it is of great significance to analyze the influence of the number of segments in the cladding on the bending performance. The parameters of the fiber are a = 30 μm, $\lambda$ = 1064 nm, b = 62.5 μm, dn = dn1 = 0.0006, R = 10 cm, $\gamma$ = 0.65, $\theta$ = 45°. The bending loss of $LP_{01}$ and $LP_{11V}$ fibers and the loss ratio influenced by different numbers of segments are shown in Figure 6a. With the number of segments increasing, the bending loss of $LP_{01}$ will decrease. When the number is 0, 2, 4, 6, 8, and 10, the bending loss of the $LP_{01}$ are 11.4687 dB/m, 0.3196 dB/m, 1.5156 dB/m, 0.1306 dB/m, 0.022177 dB/m, and 0.011 dB/m, respectively. The bending loss of $LP_{11v}$ is 120 dB/m, 58.52 dB/m, 0.092 dB/m, 9.4405 dB/m, 1.9636 dB/m, and 0.314 dB/m respectively. When N is 8, the loss ratio will be close to 90, the highest in all points. Figure 6b plots effective mode area with the number of segments in the cladding. The overall trend of effective mode area will decrease with N increasing. When N is 0 and 10, the effective mode areas are 804.35 μm$^2$ and 751.27 μm$^2$, respectively. However, the effective mode area will be without big changes, except N = 0.

Figure 6c plots electric mode lines with the number of segments in the cladding. With N increasing, the refractive index of cladding will decrease, leading to the increase of the effective refractive index difference between core and cladding. The enhancement of the optical beam ability of the fiber core results in the reduction of the loss of $LP_{01}$ and $LP_{11V}$ and the effective mode area. We suggest a PPC-SCF that consists of 8 periods of segmentation with a = 30 μm, b = 62.5 μm, dn = dn1 = 0.0006, and $\gamma$ = 0.65, $\lambda$ = 1.064 μm at R = 10 cm.

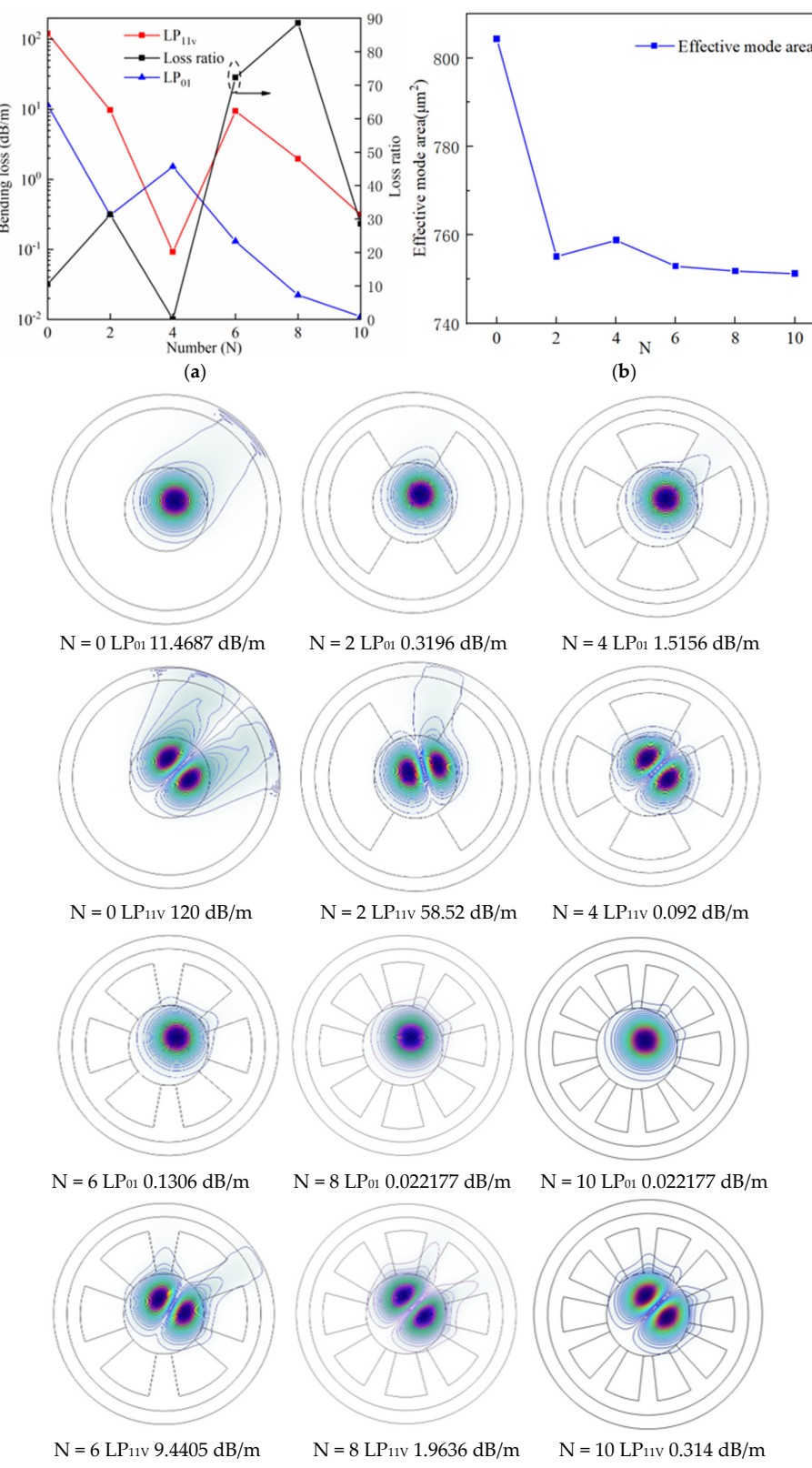

N = 0 LP$_{01}$ 11.4687 dB/m     N = 2 LP$_{01}$ 0.3196 dB/m     N = 4 LP$_{01}$ 1.5156 dB/m

N = 0 LP$_{11v}$ 120 dB/m     N = 2 LP$_{11v}$ 58.52 dB/m     N = 4 LP$_{11v}$ 0.092 dB/m

N = 6 LP$_{01}$ 0.1306 dB/m     N = 8 LP$_{01}$ 0.022177 dB/m     N = 10 LP$_{01}$ 0.022177 dB/m

N = 6 LP$_{11v}$ 9.4405 dB/m     N = 8 LP$_{11v}$ 1.9636 dB/m     N = 10 LP$_{11v}$ 0.314 dB/m

(**c**)

**Figure 6.** (**a**) the bending losses of the LP$_{01}$ and LP$_{11v}$ modes and loss ratio and (**b**) the effective mode area (**c**) the electric field lines of LP$_{01}$ and LP$_{11v}$ modes with the N.

### 3.3. Effects of Bending Angle

The influence of bending angle on the bending loss, the effective mode area, and the electric mode lines of the proposed fiber are displayed in Figure 7. Due to the asymmetric structure of the fiber, the bending performance of the proposed fiber in different orientations from 0° to 45° needs to be analyzed. The parameters are: $\lambda$ = 1064 nm, a = 30 $\mu$m, b = 62.5 $\mu$m, dn = dn1 = 0.0006, R = 10 cm, $\gamma$ = 0.65, and R = 10 cm. From Figure 7a, although the bending loss of fiber $LP_{01}$ mode (FM) firstly increases and then decreases, it remains lower than 0.1 dB/m with the bending angle varying from 0° to 45°. The lowest bending loss $LP_{11v}$ first decreases and then increases. When the bending angle is 23°, the bending loss of $LP_{11v}$ is lower than 1dB/m, which cannot be effective for single-mode operation. In a bending period, the bending angle from 13° to 32° cannot be transferred into a single mode. The bending angle range of the fiber has been greatly improved compared with N = 4 and 6. As Figure 7b shows, the effective mode area will remain unchanged with the increase of bending angle. The effective mode area is 751.85 $\mu$m$^2$.

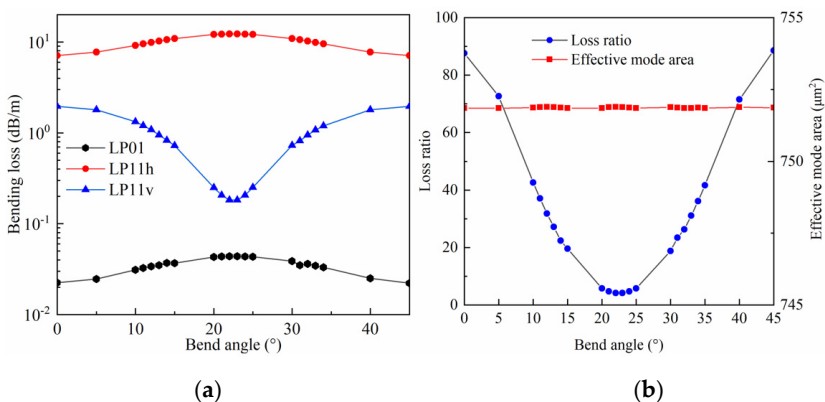

(**a**)  (**b**)

**Figure 7.** Variations of (**a**) the bending losses of the $LP_{01}$, $LP_{11v}$ and $LP_{11h}$ modes and (**b**) the effective mode area loss ratio with the bending angle.

Figure 8 shows the electric field lines with different bending angles. Combing Figures 7a and 8, we can see that when the bending angle increases to 22.5°, the $LP_{11v}$ mode is restricted to the fiber core. Compared with other bending angles, the electric field line radiated into the cladding is significantly reduced, resulting in a loss of less than 1 dB/m.

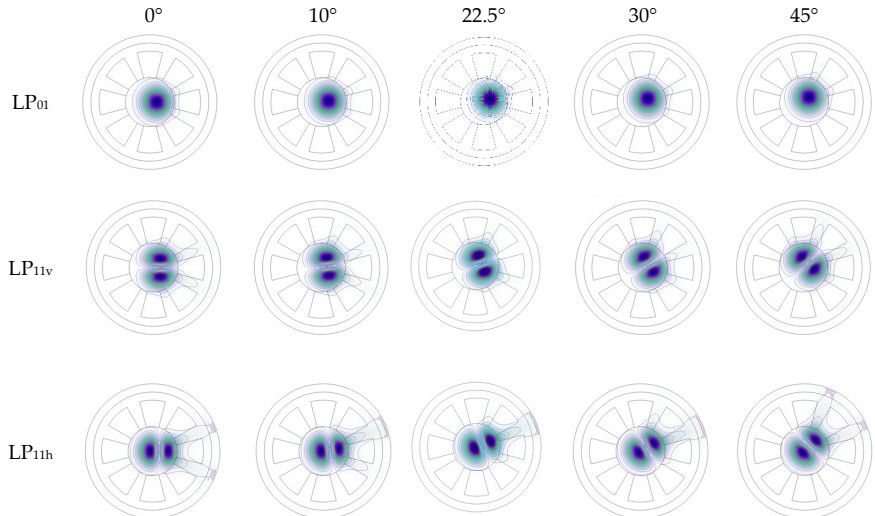

**Figure 8.** The electric field of the $LP_{01}$, $LP_{11v}$, and $LP_{11h}$ mode(s) of the fiber with the bending angle of 0°, 10°, 22.5°, 30°, 45°.

### 3.4. Effects of Duty Cycle

The influence of the duty cycle is shown in Figure 9. The parameters are: $\lambda$ = 1064 nm, a = 30 μm, b = 62.5 μm, dn = dn1 = 0.0006, R = 10 cm, and $\theta$ = 0°. Figure 9a plots the effects of $\gamma$ on the bending loss of the structure. The bending loss of $LP_{01}$, $LP_{11v}$, and $LP_{11h}$ decreases sharply with the value of $\gamma$ increasing. Because of the increasing value of $\gamma$, the index contrast between the core and cladding enhances. The proposed fiber can filter the higher order modes to realize single mode operation within 0.5 < $\gamma$ < 0.7.

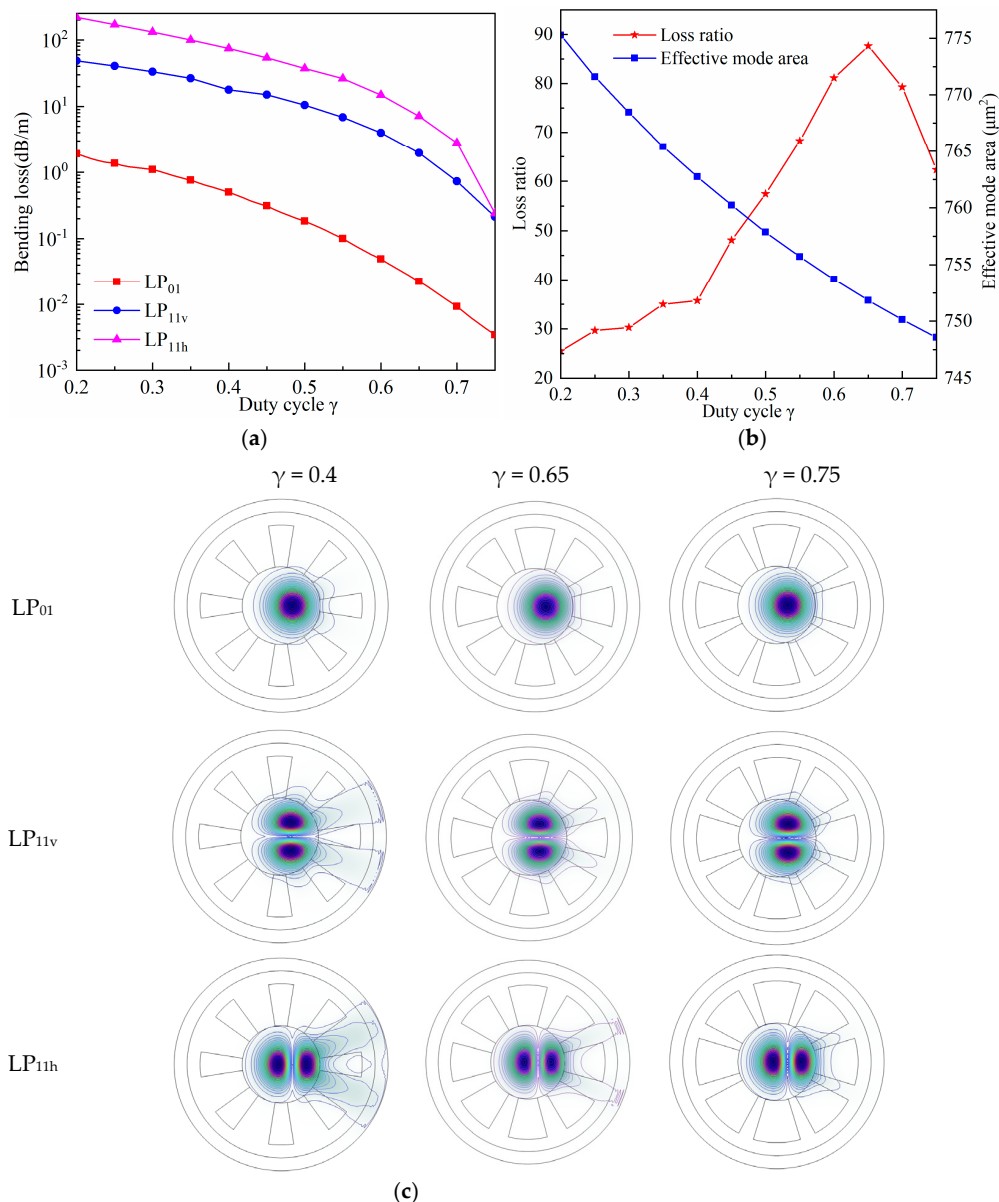

**Figure 9.** (**a**) the bending losses of the $LP_{01}$, $LP_{11v}$, and $LP_{11h}$ modes and (**b**) the effective mode area loss ratio, and (**c**) the electric field of the $LP_{01}$, $LP_{11v}$, and $LP_{11h}$ with the $\gamma$.

Figure 9b plots effective mode area and loss ratio as a function of $\gamma$. The effective mode area decreases as the value of $\gamma$ increases. With $\gamma$ ranging from 0.2 to 0.75, the effective mode area decreases from 775.21 μm² to 748.58 μm². Figure 9c shows the electric field lines with the duty cycle $\gamma$. Combing Figure 9a–c, it can be seen that with the increase of $\gamma$, the electric field line is gradually limited to the core, reducing the loss of each order mode and effective mode area.

In order to further increase the effective mode area, the core radius of the proposed is scaled to 38 μm, ensuring the cladding does not change. The parameters are: $\lambda$ = 1064 nm, a = 38 μm, b = 62.5 μm, dn = dn1 = 0.0006, R = 10 cm, $\theta$ = 0°. The bending loss of $LP_{01}$, $LP_{11v}$, and $LP_{11h}$ is 0.095 dB/m, 2.66 dB/m, and 6.98 dB/m, respectively. The proposed fiber still can realize the single mode operation. Moreover, the effective mode area can reach up to 952 μm² at a 10 cm bending radius.

Figure 10 plots the bending loss and effective mode area of $LP_{01}$, $LP_{11v}$, and $LP_{11h}$ as a function of heat load. As can be seen from Figure 10, the bending loss of $LP_{01}$, $LP_{11v}$, and $LP_{11h}$ will decrease with heat load rising. The bending loss of $LP_{01}$ and $LP_{11v}$ reduces from 0.095 dB/m to 0.031 dB/m, 2.66 dB/m to 0.98 dB/m with the heat load ranging from 0 W/m to 18 W/m, respectively. Therefore, the proposed fiber cannot realize single mode operation with heat load = 18 W/m. The effective mode area also diminishes with the heat load increasing. The effective mode field area decreased from 952.24 μm² to 929.69 μm².

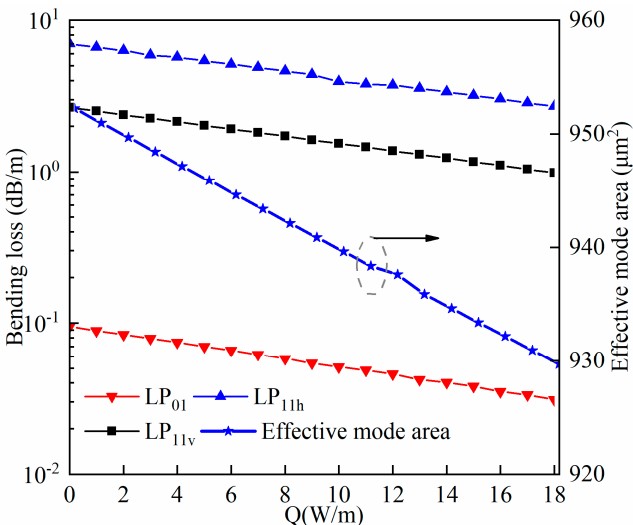

**Figure 10.** The bending loss of $LP_{01}$, $LP_{11v}$ and $LP_{11h}$ and effective mode area as function of heat load.

## 4. Conclusions

In summary, this paper proposes a novel fiber design. The proposed fiber is analyzed by the FEM. The significant advantages of the proposed fiber over step-index fiber are demonstrated. For the proposed design, the mode area can scale to 952 μm² with effective SM operation at a 10-cm bending radius. The proposed fiber achieves certain advantages of practical fabrication, which can be suitable for compact high-power lasers.

**Author Contributions:** Conceptualization and methodology, S.Y., W.Z. and Y.S.; software, validation and writing—original draft preparation, S.Y.; data analysis, H.D.; writing—review and editing Y.S.; funding acquisition, S.T. All authors have read and agreed to the published version of the manuscript.

**Funding:** This work is supported by the Guangxi Key R&D Program (No. AB22035047), Guangxi Key Research and Development Plan (No. AB21075009) and Guangxi Key Laboratory of Manufacturing System & Advanced Manufacturing Technology, Guilin University of Electronic Technology (Guilin,541004, China 22-35-4-S008).

**Institutional Review Board Statement:** Not applicable.

**Informed Consent Statement:** Not applicable.

**Data Availability Statement:** The data are available upon request.

**Conflicts of Interest:** The authors declare no conflict of interest.

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
