# Peer review of "A Large Mode Area Parabolic-Profile Core Fiber with Modified Segmented in Cladding"

_photonics, doi:10.3390/photonics9100783_

Round 1

Author Response

Dear Reviewer:

Thank you very much for the comments on the present work, we have carefully read the questions and advices in the comments and have tried our best to correct the present paper. The answers and corrections  see the attachment.

Reviewer 2 Report

See the attachment, please.

Author Response

    Thank you very much for taking the time to revise my paper. According to the comments, I have modified my paper . I have learned a lot of things, your professional suggestion deserves our deep thinking. I will pay more time to continue learning in the study to meet your expectations.  Please see the attachment.

Reviewer 3 Report

Title: A bend-resistant low bending loss and large mode area modified segmented trench in cladding fiber with parabolic-profile core

Authors: Yang Song, et al.

Comments:

The authors presented a bending-resistant fiber design with low bending loss and a large mode area, and studied the effects of structural parameters on the mode field area and bending loss. However, the novelty of this paper is not well presented, and some important information for the design consideration is not given. Therefore, I suggest this paper could be published only if the author can properly address the following questions.

Here are my concerns:

1.   In the abstract, the author claimed ‘The proposed fiber can offer an effective SM operation with an LMA of 952 µm2 … the fiber is also suitable under a 11 W/m heat load’. However, after reading through the whole manuscript, it seems that ‘the 952 um2 mode filed area’ and ‘suitability for 11W/m heat load’ are not achieved in one fiber. Please correct the statement.

2.      In the introduction part, the authors listed the ‘shortcomings’ of some specialty fibers. However, the author did not explain the advantages of the proposed fiber design compared with the aforementioned ones. Moreover, there are some fiber designs that adopt segmented trenches having been reported before; what is the improvement of the proposed fiber design? This is where the novelty of this work lies; please specify.

3.      Parameters dn and dn1 are not defined in the manuscript; please specify.

4.      How would the length of the trench (along the radial direction) affect the bending loss and the mode field area?

5.      The refractive index profile (RIP) of the core could largely affect the fiber performance. Could the author discuss how the parabolic parameter of the core RIP affects the fiber performance, and why the authors choose Equation (1) to model the RIP?

6.      There are some inconsistencies in the manuscript. For instance, the wavelength is specified to be 1550 nm when calculating the bending loss (Page 3). However, the wavelength is 1.064 um on Page 2 and 1.06 um on Page 4. Please check carefully to avoid such kind of problems.

7.      The author mentioned ‘leakage loss’ on Page 6. How is it distinguished from ‘bending loss’ in this manuscript?

8.      On Page 13, the author concluded ‘0. 5 < γ < 0.7 would be appropriate values of γ for filtering the higher order modes’. What are the standards for ‘appropriate values’? Such description is too vague, and the author should make it clear.

9.      The value of some parameters, such as β, ksi, kac, h, in Equation 5 are not given when studying the effects of heat load; please specify. And what are the structural parameters of the fiber in section E?

10.   The authors studied the effects of some parameters on the performance of the proposed fiber design. However, it seems the author obtained the final result simply by scaling the core diameter to 38 um without optimizing the parameters. I can not help to wonder what is the basis for choosing this diameter. Is this the best parameter? Could the core size be further scaled for an even larger mode area? Please provide more detailed information for designing the final parameter of the fiber.

11.   There are many typo even logical problems, just too many for me to make a full list here. Please proofread the manuscript carefully.

a)       With the core radius increasing from 24 to 35, the bending loss of LP01 decreases from 0.0136dB/m to 0.052dB/m. (logic problem)

b)       The Proposed fiber can realize the single mode operation compared with step-core fiber at the same bending radius. (typo error)

c)       ‘Also, large mode field area fibers must work in a single operation to ensure beam propagation quality in high-power lasers. ’ (Do you mean single-mode operation?)

d)       ‘However, the challenge is that increasing the mode field area of the fiber while maintaining single-mode operation can affect beam quality.’ (logic problem)

e)      

Author Response

   Thank you very much for taking the time to revise my paper. According to the comments,  I have modified my paper. I also learned a lot of things, your professional suggestion deserves our deep thinking. I will pay more time to continue learning in the study to meet your expectations.Please see the attachment.

Round 2

Reviewer 3 Report

I am satisfied with the response and the revised manuscript.

Author Response

Dear Reviewer:

Thank you very much for your professional suggestion. According to your suggestion, the paper has been carefully revised. The revised section clearly will be shown highlight in revised paper. In the future, I will pay more attention to writing.

Thank you very much for taking the time to revise my paper. According to the comments, I also learned a lot of things. I will pay more time to continue learning and improve the expression of the paper to meet your expectations.
